# Quality Assessment of Red Wine Grapes through NIR Spectroscopy

Maria Inês Rouxinol [1], Maria Rosário Martins [2,*], Gabriela Carneiro Murta [1], João Mota Barroso [3] and Ana Elisa Rato [3,*]

[1] MED—Mediterranean Institute for Agriculture, Environment and Development, IIFA—Institute for Advanced Studies and Research, Universidade de Évora, Pólo da Mitra, Ap. 94, 7006-554 Evora, Portugal; mir@uevora.pt (M.I.R.); gabriela.murta@gmail.com (G.C.M.)

[2] HERCULES Laboratory, Departamento de Ciências Médicas e da Saúde, Escola de Saúde e Desenvolvimento Humano, Universidade de Évora, Rua Romão Ramalho 59, 7000-671 Evora, Portugal

[3] Departamento de Fitotecnia, Escola de Ciências e Tecnologia, Universidade de Évora, Pólo da Mitra, Ap. 94, 7006-554 Evora, Portugal; jmmb@uevora.pt

* Correspondence: mrm@uevora.pt (M.R.M.); aerato@uevora.pt (A.E.R.)

**Abstract:** Red wine grapes require a constant follow-up through analytical chemistry to assure the greatest wine quality. Wet chemical procedures are time-consuming and produce residues that are hard to eliminate. NIR (near infrared radiation) spectroscopy has been referred as an accurate, rapid, and cost-efficient technique to evaluate quality in many fruit species, both in field and in industry. The main objective of this study was to develop predictive models using NIR spectroscopy to quantify important quality attributes in wine grapes. Soluble solids content (SSC), titratable acidity (TA), total phenolic content, total flavonoids, total anthocyanins, and total tannins were quantified in four red wine grape varieties, 'Aragonês', 'Trincadeira', 'Touriga Nacional', and 'Syrah'. Samples were collected during 2017 and 2018 along véraison. Prediction models were developed using a near-infrared portable device (Brimrose, Luminar 5030), and spectra were collected from entire grapes under near field conditions. Models were built using a partial least square regression (PLSR) algorithm and SSC, TA, total anthocyanins, and total tannins exhibited a determination coefficient of 0.89, 0.90, 0.87, and 0.88, respectively. The Residual Prediction Deviation (*RPD*) values of these models were higher than 2.3. The prediction models for SSC, TA, total anthocyanins, and total tannins have considerable potential to quantify these attributes in wine grapes. Total flavonoids and total phenolic content were predicted with a slightly lower capacity, with $R^2$ = 0.72 and 0.71, respectively, and both with a RPD of 1.6, indicating a very low to borderline potential for quantitative predictions in flavonoids and phenols models.

**Keywords:** NIR-spectroscopy; phenolic; flavonoids; anthocyanins; tannins; SSC; wine grapes





## 1. Introduction

In modern viticulture, it is necessary to monitor the evolution of many quality parameters to adopt the best selective harvesting management and to produce high quality wines. Most of the phenolic compounds quantified in wine are derived primarily from the grape berry skin, and the most representative compounds in berry skin and seeds are flavonoids, anthocyanins, and tannins [1].

Phenolic compounds in wine determine many of its sensory qualities, since they are related to colour, flavour, and taste, which make it essential to monitor these compounds in the vineyard during ripening. Phenolic compounds usually bond by intermolecular interactions to volatile compounds that influence wine aroma [2]. Wine has, in its constitution, a great part of phenolic compounds, which justifies the extreme importance of their determination during grape maturation [3]. Consumers are becoming even more demanding in terms of wine quality and curious about the product that they are consuming [4,5]. Thus, it

becomes essential to find techniques that allow the quick determination of wine quality parameters. Sugar content, acidity, total polyphenols, total flavonoids, total anthocyanins, and total tannins are considered as key components of wine quality [6,7].

Phenolic compounds present in red grapes and red wine, known for their health benefits, are important antioxidants and chemoprotectant agents against cancer (such as colon or breast cancer) and other degenerative diseases often associated with oxidative and inflammatory process (e.g., Parkinson and Alzheimer diseases), inhibition of platelet aggregation, and antidiabetic potential as well as anti-allergenic, anti-ulcer, and antimicrobial properties [8–11]. Flavonoids have been catching interest since they showed strong antioxidant capacity [12–14] and also might have an impact on wine organoleptic characteristics, namely colour and aroma [2,15]. Anthocyanins are flavonoids responsible for the colours red, blue, and purple in plant tissues [16], and in red wine grapes, these compounds are essentially found on skins [17]. Malvidin-3-O-glucoside is the most abundant anthocyanin in grapes and wine, representing about 40% of their content [4]. In red grapes, anthocyanins constitute the largest percentage of phenolic compounds, being an important constituent highlighting the sensorial qualities of the wine [18]. The wine colour is a very important quality parameter and depends on the content of phenolic compounds in grapes, as well as on the oenological and storage conditions [19,20]. Furthermore, tannins play an important role in the degustation, since, together with flavan-3-Ols and proanthocyanidins, they contribute to the wine's body and mouth feel [21].

Nevertheless, phenolic compounds' determination and quantification can be tricky due to their complexity and structural diversity. Many analytical methods can be used to quantify the phenolic compounds or anthocyanins' content in plant extracts [22], such as Folin–Ciocalteu for total phenol content, Folin–Denis for tannins quantification, or differential pH methods to evaluate anthocyanins' content. Even though the Folin–Ciocalteu method and the differential pH method are recognized as reference methods for grape analysis due to their simplicity and low cost, in complex matrices such as grapes extracts, these methods have a lack of specificity frequently leading to overestimations [23].

Despite the existence of recommended methods to quantify phenolic compounds, the diversity of protocols makes it difficult to select the best method to be employed [24]. Additionally, the purification, separation, quantification and identification of anthocyanins and phenols have been widely studied using High Performance Liquid Chromatography (HPLC) coupled with a Diode Array Detector (DAD) [25–30], however, these procedures depend on specific equipment and specialized and skilled labour.

Despite the great impact of phenolic compounds in wine quality, the balance between the sugar content and acidity builds up the grape flavour, that will also influence wine quality. If acidity is low, grape musts and wines can lose their organoleptic properties, while high acidity is an undesirable characteristic [31]. Grape sugar content has a great influence on wine quality due to the transformation of sugar in alcohol through alcoholic fermentation. The acidity and sugar accumulation in grape berries is a complex process which depends on many environmental factors, hence the quantification of these two groups during grape ripening is imperative in wine production as a guarantee of quality [5].

In modern viticulture, there is a need of eco-friendly methods that are easy to execute and adapt to field conditions. During the last 20 years, there has been an increasing interest in using spectroscopic techniques such as NIR due to the many advantages of these techniques. NIR spectroscopy has been referred as an accurate, rapid, and cost-efficient technique to evaluate quality parameters in many fruit species, both in the field and in the industry [32]. In wine and grape must the NIR spectroscopy has been used to predict phenols [33], tannins [34], SSC, and anthocyanins [35] with determination coefficients over 0.9. Although these results have proved the feasibility of the NIR spectroscopy for quality control, this technique is not widely used in field conditions. Santos Costa et al. [35] have developed prediction models for flavonoids to be used in the field, with a determination coefficient of 0.7. However, the NIR spectra were collected in a dark chamber, which may limit the application of these models to natural conditions.

The use of NIR spectroscopy on intact grapes in the vineyards would enable the assessment of various quality parameters simultaneously in the fields and a more efficient decision-making process in the winery. In NIR spectroscopy, the spectra information is based on the overtones and combination bands of fundamental molecular vibrations (namely C-H, N-H, O-H, and S-H) observed in the mid-IR spectral region. Chemical bonds present in complex matrixes vibrate at specific frequencies. These vibrations are determined by the mass of the atoms, the shape of the molecule, the stiffness of the bonds, and the periods of associated vibrational coupling [36]. Therefore, most biochemical and chemical species have specific absorption bands in NIR spectra regions that can be used for identification and quantification due to the vibrations of the atoms on the molecules when these are irradiated with NIR frequencies [36].

The analysis and interpretation of the NIR spectra need proper chemometric tools that combine mathematical and statistical methods to extract relevant information from multivariate chemical data such as NIR spectra. Despite the huge potential of multivariate methods in general, the Partial Least Square Regression has been extensively applied to build multivariate classification models [37]. This involves the development of mathematical models to correlate the presence of the analyte and the instrumental responses obtained by the determination of samples that contain the analyte in known concentrations [37]. The PLS regression is based on assuming that only the combined use of determined wavelengths can generate true concentration values. These wavelength sets are compressed to factors and then the lower number of factors is selected based on the lower cross-validation error [38].

The main goal of this study is the development of new methodologies using NIR spectroscopy to assess the most important quality parameters in the grapes that will influence the final quality of the wine.

## 2. Materials and Methods

### 2.1. Grape Sampling

'Aragonês', 'Syrah', 'Touriga Nacional' and 'Trincadeira' (*Vitis vinifera* L.) grapes were sampled during grape ripening in two consecutive years, 2017 and 2018, from a vineyard located in Herdade da Mitra, Valverde, Évora (centre/south Portugal) (38°32′01.2″ N 8°00′57.7″ W) (Figure 1). To ensure an appropriate variability, 100 plants were selected from each variety and 25 clusters were picked randomly on each sampling date, from different raws and different plants, avoiding collecting samples from the same plant twice. Samples were harvested weekly, starting at véraison and continuing until commercial harvest.

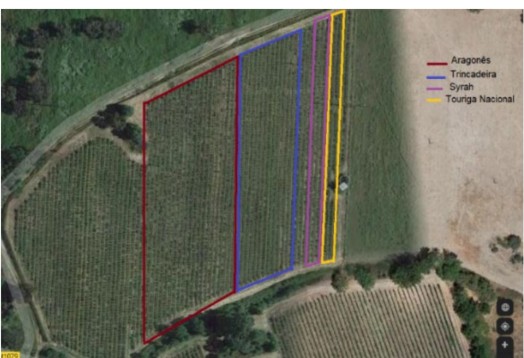

**Figure 1.** Herdade da Mitra Vineyard (38.533567, −8.015437).

### 2.2. Grape Skin Extracts

Berries were manually separated from clusters, and 3 replicates of 50 grape berries each were selected, weighed (Figure 2), and frozen at −18°C until manual separation of skin, pulp, and seeds. These fractions were weighted, and the skin fraction was frozen for further analysis. The pulp was used for quantifying TA and SSC, and these parameters were

determined by refractometry and potentiometry, respectively; both results were expressed as %.

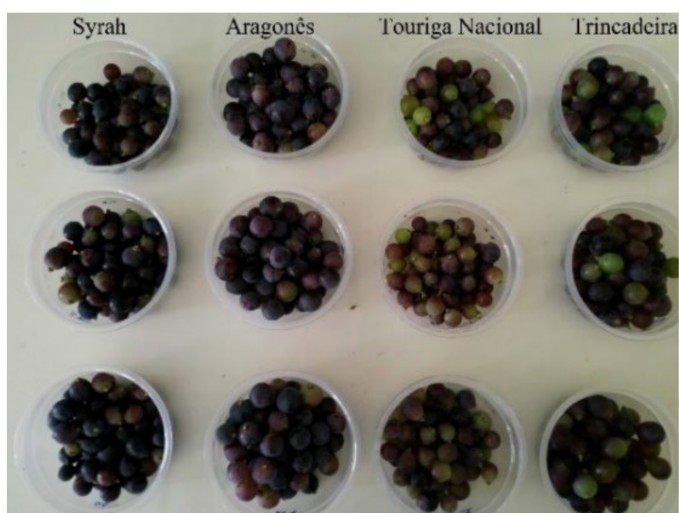

**Figure 2.** Berries collected at the beginning of vérasion in 2018.

To prepare grape skin extracts, 10 g of skin fraction was added to 25 mL of extraction reagent (ethanol:water:hydrochloric acid 37%) and ground using a IKA T25 digital Ultra-Turrax at 9000 rpm for 30 s. Samples were extracted in agitation for one hour, filtered, and stored at −20 °C for further analysis.

*2.3. Reference Analysis*

2.3.1. Total Phenolic Content

The total phenolic content, which represents the bulk of phenols in grape skin extracts, was determined using an adaptation of the method described by Gajula (2009) [39] to microplate. Gallic acid was used as an analytical standard with a concentration range between 10 and 500 μg/mL, and the blank reference was pre-formed with the extraction reagent. Six replicates were performed for both samples and standards. After the plates' incubation at 40 °C for 30 min, the absorbance was read at 630 nm. Total phenolic values were expressed in μg of gallic acid equivalents by ml of grape skin extract.

2.3.2. Total Flavonoids

Total flavonoids content in grape extracts was quantified according to an adaptation of the method described by Hosu (2013) [40] to microplate. Briefly, Rutin was used as an analytical standard with a concentration range between 0 and 200 μg/mL. Six replicates were performed for both samples and standards, samples were diluted with water, and then $AlCl_3H_2O$ 2% and $C_2H_3NaO_2$ was added. After incubating for 15 min at room temperature, the absorbance was measured at 430 nm. Results were expressed in μg of Rutin equivalent by ml of grape skin extract.

2.3.3. Total Anthocyanins

Total anthocyanins were quantified by chromatography using a modification of the method by Antoniolli (2015) [41]. For chromatographic analysis, an UPLC Dionex Ultimate 3000 was used, equipped with a diode array detector (DAD) and Chromeleon 6.8 software. Anthocyanin identification was performed according to Soriano et al. (2007) [35]. The quantification was conducted at a wavelength of 520 nm and the anthocyanin content expressed in mg Oenin-3-O-glucoside equivalents by ml of grape skin extract after using a calibration curve with the concentration range of 0.0125–0.1 mg/mL and a $R^2$ = 0.9998.

### 2.3.4. Total Tannins

Total tannins were determined according to the modification of the colorimetric method, Folin–Denis, by Singleton (1998) [42] and adapted to microplate. This method is based on the reduction in phosphomolybdic acid and tungstic in alkaline medium. When in the presence of tannins, it develops a blue colour, measured at 760 nm [43]. The absorbance was read at 760 nm after 30 min of incubation; both samples and standards were performed in sextuplicate. A calibration curve was prepared at a concentration range between 0 and 1.5 mg/mL. The results are expressed in mg of tannic acid equivalent by ml of grape skin extract.

### 2.4. Spectra Collection

In each time point and in intact grapes, sample spectra and measurements were taken by scanning the 3 replicates of 50 berries (Figure 2) using a Brimrose Luminar 5030 AOTF-NIR spectrometer equipped with an indium–gallium–arsenide (InGaAs) detector, using Acousto Optical Tunable Filter technology and a probe of 6 mm of sampling area. Spectra were collected across the wavelength range of 1100–2300 nm with an increment wavelength of 1 nm, in reflectance mode, and exported as Log(1/R). Spectra were collected replicating the field procedure, which means that the spectrometer collected the spectra above the sample at a sample distance of 40 mm, since there is no need of blocking the ambient light with this kind of equipment. For each sample, 250 scans were made with a spectral resolution of 2 nm and with 1200 data points per spectrum. Each sample spectrum resulted from the average of 250 scans. Samples were kept at room temperature before scanning.

### 2.5. Statistical Analysis

Spectral data and measurements were exported to Unscrambler software (version 10.4, CAMO, ASA, Oslo, Norway). The PLSR algorithm was used to obtain the models to quantify, simultaneously, different quality parameters in the grapes.

In the developed models, there are essential statistical parameters used to characterize the quality of the final models. They are: $R^2$cal (coefficient of determination in calibration), *RMSE* (Root Mean Squared Error), $R^2$val (Coefficient of Correlation in Validation) *Bias*, and *RPD* [35,40]. $R^2$cal estimates the percentage of variance on the training set that can be explained by the new model, also called 'explained variance'. It measures the ability of the model to fit the data, although this parameter alone is not conclusive. Models with $R^2$ close to 1, which can be found in small datasets, usually do not have a good capacity for prevision. Lower $R^2$cal values mean that there is a low reproducibility and the information from the spectra cannot explain the chemistry of the samples [44]. $R^2$val corresponds to the percentage of variance in the validation set explained by model [44].

*RMSE* indicates the error in which a sample can be predicted and is comparable to a *RMSECV* when a cross-validation procedure is used instead of using a separate sample test.

$$RMSE = \sqrt{\frac{\sum_{i=1}^{N}(y_i - x_i)2}{N}} \qquad (1)$$

Both values are expressed in the same units as the reference values and are strongly influenced by the analyte concentration [40]. $N$ is the sample number, $y_i$ the reference value of sample $i$, and $x_i$ is the predicted value for the sample $i$ [38]. *RPD* value is another quality parameter that includes the variance of the reference values in the quality evaluation of the model [44]. To avoid over optimistic evaluations with a relatively small range of reference values, the *RPD* is calculated by the following equation [44]:

$$RPD = \frac{\text{SD}}{RMSE} \qquad (2)$$

*RPD* is the quotient of the standard deviation (SD) of the reference values and the corrected mean error of the validation. For instance, if the *RMSE* value is high compared

with the variance of the analyte's concentration in all samples, the obtained *RPD* is low, which means that the model cannot accurately predict a sample's variability.

*Bias* is an indicator of the systematic error in the predictive values and is calculated as the average deviation between the reference values and the predicted values [38].

$$Bias = \frac{\sum_{i=1}^{N}(x_i - y_i)}{N} \tag{3}$$

where $N$ is the number of samples, $x_i$ the predicted value for the $i$ sample, and $y_i$ the reference value for the $i$ sample [38].

The models were validated using a cross-validation procedure due to the reduced number of samples, less than 100. In the cross-validation procedure, one sample from the calibration dataset is removed and a model is set up with the remaining samples. The error for the analysis for this sample is calculated. The process is repeated until all the samples of the calibration dataset have been analysed.

The multivariate calibration models were developed using Partial Least Squares Algorithm and different spectra pre-treatments.

## 3. Results

### 3.1. Chemical Parameters

Table 1 shows the mean and the SD of the SSC and TA obtained in both years from véraison until harvest. In all grape varieties, SSCs were higher in 2017 than in 2018. TA also showed differences between years, being lower in 2017. During the sampling period, SSC increased while TA decreased, and the evolution of both parameters was expected during this period. Despite the differences between years, these values are characteristic for Alentejo vineyards for these varieties.

**Table 1.** Mean ± standard deviation (SD) of SSC and TA in grape berries from samples collected in 2017 and 2018 in the days after véraison (dav).

| Variety | Dav | 2017 | | 2018 | |
|---|---|---|---|---|---|
| | | SSC (%) | TA (%) | SSC (%) | TA (%) |
| 'Syrah' | 0 | 12.57 ± 0.68 | 0.789 ± 0.035 | 12.50 ± 0.08 | 1.963 ± 0.004 |
| | 7 | 16.30 ± 0.00 | 0.519 ± 0.020 | 14.40 ± 0.14 | 1.633 ± 0.238 |
| | 15 | 20.73 ± 0.17 | 0.353 ± 0.025 | 18.45 ± 0.04 | 0.994 ± 0.015 |
| | 21 | 25.10 ± 0.08 | 0.288 ± 0.001 | 19.67 ± 0.12 | 0.755 ± 0.020 |
| | 30 | 27.73 ± 0.09 | 0.271 ± 0.013 | 19.97 ± 0.29 | 0.696 ± 0.019 |
| 'Aragonês' | 0 | 15.47 ± 0.21 | 0.514 ± 0.025 | 14.67 ± 0.05 | 1.223 ± 0.005 |
| | 7 | 16.93 ± 0.05 | 0.361 ± 0.028 | 16.13 ± 0.09 | 0.826 ± 0.010 |
| | 15 | 19.80 ± 0.16 | 0.286 ± 0.002 | 20.70 ± 0.08 | 0.625 ± 0.001 |
| | 21 | 21.90 ± 0.41 | 0.289 ± 0.001 | 21.77 ± 0.17 | 0.575 ± 0.014 |
| | 30 | 23.87 ± 0.54 | 0.197 ± 0.006 | 21.43 ± 0.37 | 0.485 ± 0.001 |
| 'Trincadeira' | 0 | 12.97 ± 0.05 | 0.768 ± 0.020 | 11.53 ± 0.45 | 2.121 ± 0.007 |
| | 7 | 14.77 ± 0.48 | 0.517 ± 0.016 | 12.83 ± 0.09 | 1.750 ± 0.093 |
| | 15 | 20.10 ± 0.22 | 0.332 ± 0.024 | 17.05 ± 0.04 | 0.863 ± 0.010 |
| | 21 | 21.23 ± 0.48 | 0.284 ± 0.001 | 18.53 ± 0.34 | 0.732 ± 0.013 |
| | 30 | 25.30 ± 0.70 | 0.283 ± 0.003 | 20.13 ± 0.12 | 0.588 ± 0.006 |
| 'Touriga Nacional' | 0 | 10.42 ± 0.31 | 0.915 ± 0.013 | 8.13 ± 0.29 | 2.643 ± 0.005 |
| | 7 | 14.10 ± 0.08 | 0.578 ± 0.004 | 13.00 ± 0.08 | 1.845 ± 0.116 |
| | 15 | 18.47 ± 0.73 | 0.345 ± 0.025 | 16.35 ± 0.04 | 1.063 ± 0.010 |
| | 21 | 20.50 ± 0.16 | 0.286 ± 0.005 | 17.93 ± 0.09 | 0.817 ± 0.002 |
| | 30 | 22.70 ± 0.51 | 0.252 ± 0.002 | 18.67 ± 0.12 | 0.683 ± 0.041 |

Table 2 shows the total phenolic content, total flavonoids, total anthocyanins, and total tannins extracted from berry skins in all studied varieties. The total phenolic content, anthocyanins, flavonoids, and tannins during the sampling period ranged from 0.19 ± 0.07

to 0.92 ± 0.06 (mg Gallic acid eq/mL grape skin extract), 0.02 ± 0.01 to 0.17 ± 0.01 (mg Rutin eq/mL grape skin extract), 0.18 to 4.25 (mg Oenin-3-O-glucosides/mL grape skin extract), and 0.25 ± 0.07 to 5.57 ± 0.09 (mg Tannic acid eq/mL grape skin extract), respectively. These ranges of values are characteristic of these varieties produced in the Alentejo region, which means that they incorporate the expected variability which ensures the representativeness of the samples.

**Table 2.** Total of phenolic content (mg Gallic acid equivalent/mL of extract), flavonoids (mg Rutin eq/mL grape skin extract), anthocyanins (mg Oenin-3-O-glucosides/mL of grape skin extract), and tannins (mg Tannic acid eq/mL grape skin extract) extracted from grape skin from samples collected in 2017 and 2018 in the days after véraison (dav).

| Variety | Year | Dav | Total Phenolic (mg Gallic Acid eq/mL) | | Total Flavonoids (mg Rutin eq/mL) | | Total Anthocyanins (mg Oenin-3-O-glucosides/mL) | | Total Tannins (mg Tannic Acid eq/mL) | |
|---|---|---|---|---|---|---|---|---|---|---|
| | | | Mean ± SD | Range | Mean ± SD | Range | Mean ± SD | Range | Mean ± SD | Range |
| 'Syrah' | 2017 | 0 | 0.39 ± 0.02 | 0.37–0.41 | 0.04 ± 0.01 | 0.02–0.05 | 0.58 ± 0.22 | 0.33–0.73 | 0.85 ± 0.08 | 0.75–0.91 |
| | | 7 | 0.43 ± 0.08 | 0.34–0.49 | 0.09 ± 0.02 | 0.07–0.11 | 1.80 ± 0.72 | 1.08–2.53 | 1.37 ± 0.15 | 1.20–0.50 |
| | | 15 | 0.61 ± 0.11 | 0.49–0.73 | 0.08 ± 0.02 | 0.05–0.09 | 2.88 ± 0.14 | 2.78–3.03 | 1.51 ± 0.06 | 1.44–1.56 |
| | | 21 | 0.63 ± 0.15 | 0.55–0.81 | 0.14 ± 0.01 | 0.13–0.15 | 3.05 ± 0.08 | 2.96–3.12 | 1.96 ± 0.09 | 1.90–2.06 |
| | | 30 | 0.59 ± 0.02 | 0.59–0.60 | 0.17 ± 0.01 | 0.17–0.17 | 2.10 ± 1.35 | 0.59–3.18 | 1.56 ± 0.03 | 1.53–1.60 |
| | 2018 | 0 | 0.38 ± 0.16 | 0.26–0.56 | 0.04 ± 0.01 | 0.04–0.05 | 0.36 ± 0.01 | 0.36–0.36 | 0.28 ± 0.03 | 0.26–0.32 |
| | | 7 | 0.37 ± 0.08 | 0.28–0.45 | 0.06 ± 0.01 | 0.05–0.06 | 1.08 ± 0.16 | 0.90–1.19 | 0.43 ± 0.14 | 0.28–0.54 |
| | | 15 | 0.59 ± 0.11 | 0.52–0.71 | 0.08 ± 0.01 | 0.07–0.09 | 1.44 ± 0.14 | 1.34–1.60 | 0.70 ± 0.05 | 0.66–0.75 |
| | | 21 | 0.46 ± 0.13 | 0.35–0.60 | 0.06 ± 0.01 | 0.06–0.07 | 1.20 ± 0.68 | 0.43–1.73 | 0.58 ± 0.03 | 0.55–0.61 |
| | | 30 | 0.57 ± 0.04 | 0.56–0.62 | 0.05 ± 0.01 | 0.05–0.06 | 1.69 ± 0.37 | 1.28–2.02 | 0.55 ± 0.01 | 0.54–0.56 |
| 'Aragonês' | 2017 | 0 | 0.52 ± 0.01 | 0.51–0.52 | 0.03 ± 0.01 | 0.02–0.03 | 1.12 ± 0.19 | 0.94–1.33 | 2.46 ± 0.02 | 2.44–2.49 |
| | | 7 | 0.50 ± 0.03 | 0.47–0.52 | 0.03 ± 0.01 | 0.02–0.03 | 1.72 ± 0.15 | 1.56–1.86 | 1.63 ± 0.03 | 1.60–1.67 |
| | | 15 | 0.51 ± 0.05 | 0.46–0.55 | 0.04 ± 0.01 | 0.03–0.05 | 2.93 ± 1.11 | 2.12–4.19 | 5.45 ± 0.09 | 5.39–5.55 |
| | | 21 | 0.42 ± 0.07 | 0.35–0.49 | 0.04 ± 0.01 | 0.03–0.04 | 3.14 ± 0.39 | 2.91–3.60 | 5.57 ± 0.09 | 5.48–5.66 |
| | | 30 | 0.53 ± 0.01 | 0.52–0.55 | 0.04 ± 0.01 | 0.04–0.04 | 2.62 ± 0.42 | 2.15–3.00 | 1.29 ± 0.11 | 1.18–1.41 |
| | 2018 | 0 | 0.41 ± 0.13 | 0.26–0.51 | 0.08 ± 0.01 | 0.07–0.08 | 0.29 ± 0.07 | 0.25–0.37 | 0.62 ± 0.14 | 0.52–0.78 |
| | | 7 | 0.50 ± 0.06 | 0.44–0.56 | 0.07 ± 0.01 | 0.07–0.08 | 1.10 ± 0.31 | 0.78–1.40 | 0.56 ± 0.08 | 0.51–0.65 |
| | | 15 | 0.49 ± 0.06 | 0.45–0.56 | 0.08 ± 0.01 | 0.08–0.09 | 1.09 ± 0.06 | 1.05–1.17 | 0.86 ± 0.06 | 0.81–0.93 |
| | | 21 | 0.38 ± 0.10 | 0.30–0.49 | 0.09 ± 0.01 | 0.08–0.09 | 1.41 ± 0.13 | 1.25–1.49 | 0.41 ± 0.02 | 0.40–0.43 |
| | | 30 | 0.48 ± 0.04 | 0.44–0.52 | 0.13 ± 0.03 | 0.10–0.16 | 1.25 ± 0.07 | 1.17–1.32 | 1.32 ± 0.45 | 0.81–1.61 |
| 'Trincadeira' | 2017 | 0 | 0.33 ± 0.08 | 0.25–0.41 | 0.02 ± 0.01 | 0.01–0.03 | 0.25 ± 0.06 | 0.19–0.31 | 1.58 ± 0.31 | 1.39–1.95 |
| | | 7 | 0.31 ± 0.12 | 0.18–0.41 | 0.04 ± 0.02 | 0.02–0.06 | 1.39 ± 0.22 | 1.19–1.64 | 1.59 ± 0.03 | 1.56–1.62 |
| | | 15 | 0.25 ± 0.07 | 0.19–0.32 | 0.03 ± 0.01 | 0.03–0.05 | 1.35 ± 0.26 | 1.10–1.62 | 0.8 ± 0.06 | 0.77–0.88 |
| | | 21 | 0.46 ± 0.08 | 0.36–0.52 | 0.11 ± 0.01 | 0.10–0.11 | 2.54 ± 0.17 | 2.44–2.73 | 1.74 ± 0.11 | 1.66–1.87 |
| | | 30 | 0.51 ± 0.07 | 0.47–0.59 | 0.14 ± 0.01 | 0.14–0.15 | 3.12 ± 0.58 | 2.46–3.48 | 1.34 ± 0.12 | 1.21–1.47 |
| | 2018 | 0 | 0.19 ± 0.07 | 0.11–0.24 | 0.05 ± 0.01 | 0.04–0.06 | 0.30 ± 0.06 | 0.25–0.36 | 0.32 ± 0.13 | 0.17–0.40 |
| | | 7 | 0.24 ± 0.01 | 0.22–0.25 | 0.06 ± 0.01 | 0.05–0.06 | 0.45 ± 0.02 | 0.43–0.48 | 0.25 ± 0.07 | 0.17–0.32 |
| | | 15 | 0.41 ± 0.19 | 0.28–0.63 | 0.06 ± 0.01 | 0.05–0.07 | 0.87 ± 0.14 | 0.78–1.04 | 0.47 ± 0.12 | 0.34–0.57 |
| | | 21 | 0.44 ± 0.12 | 0.31–0.56 | 0.06 ± 0.01 | 0.05–0.07 | 0.79 ± 0.17 | 0.64–0.97 | 0.67 ± 0.04 | 0.65–0.72 |
| | | 30 | 0.37 ± 0.18 | 0.26–0.59 | 0.06 ± 0.04 | 0.06–0.06 | 0.95 ± 0.18 | 0.78–1.14 | 0.27 ± 0.12 | 0.16–0.40 |
| 'Touriga Nacional' | 2017 | 0 | 0.66 ± 0.02 | 0.65–0.68 | 0.09 ± 0.01 | 0.09–0.10 | 0.41 ± 0.03 | 0.38–0.42 | 2.17 ± 0.17 | 2.07–2.39 |
| | | 7 | 0.61 ± 0.09 | 0.50–0.69 | 0.02 ± 0.01 | 0.02–0.03 | 1.13 ± 0.25 | 0.84–1.31 | 1.28 ± 0.02 | 1.27–1.31 |
| | | 15 | 0.60 ± 0.02 | 0.59–0.63 | 0.07 ± 0.01 | 0.06–0.09 | 2.53 ± 0.36 | 2.14–2.83 | 1.30 ± 0.05 | 1.26–1.36 |
| | | 21 | 0.69 ± 0.08 | 0.60–0.74 | 0.08 ± 0.02 | 0.06–0.08 | 3.70 ± 0.60 | 3.16–4.35 | 2.13 ± 0.56 | 1.49–2.49 |
| | | 30 | 0.92 ± 0.06 | 0.86–0.99 | 0.15 ± 0.01 | 0.14–0.16 | 4.25 ± 0.48 | 3.72–4.64 | 1.52 ± 0.03 | 1.49–1.56 |
| | 2018 | 0 | 0.51 ± 0.07 | 0.47–0.59 | 0.03 ± 0.01 | 0.02–0.03 | 0.18 ± 0.11 | 0.10–0.30 | 0.60 ± 0.18 | 0.42–0.79 |
| | | 7 | 0.69 ± 0.08 | 0.61–0.78 | 0.06 ± 0.01 | 0.05–0.06 | 0.35 ± 0.06 | 0.28–0.39 | 0.80 ± 0.13 | 0.67–0.93 |
| | | 15 | 0.81 ± 0.14 | 0.67–0.94 | 0.07 ± 0.01 | 0.07–0.07 | 1.72 ± 0.25 | 1.47–1.96 | 0.94 ± 0.03 | 0.91–0.97 |
| | | 21 | 0.44 ± 0.01 | 0.44–0.45 | 0.16 ± 0.02 | 0.14–0.18 | 3.15 ± 0.07 | 3.09–3.23 | 0.81 ± 0.07 | 0.75–0.88 |
| | | 30 | 0.71 ± 0.01 | 0.70–0.72 | 0.13 ± 0.02 | 0.10–0.15 | 2.73 ± 0.25 | 2.49–2.98 | 0.83 ± 0.02 | 0.82–0.86 |

Comparing between the years, 2017 showed significantly ($p < 0.05$) higher values for phenols, anthocyanins, SSCs, and tannins ($p < 0.05$), but 2018 showed a significantly higher TA value ($p < 0.05$), and no differences were observed in flavonoids between years (Supplementary Files). Among grape varieties, 'Touriga Nacional' was the variety with significantly higher values ($p < 0.05$) in phenols, anthocyanins, and flavonoids ($p < 0.05$), and 'Trincadeira' showed the lowest values in phenols, anthocyanins, and flavonoids. 'Aragonês' was the variety with the highest tannins content and the lowest value for TA ($p < 0.05$) (Supplementary Materials).

*3.2. NIR Spectroscopy*

Figure 3 represents the NIR spectra collected from entire grapes using a Brimrose spectrometer in the wavelength range of 1100–2300 nm. Different spectral pre-processing methodologies were applied to the raw spectra to avoid irrelevant information and to obtain good prediction models. In this study, the spectra were transformed applying a first derivative procedure and the standard normal variate procedure, and the best regression model was selected for each compound. The two broad absorption bands visible in the spectra correspond to the near infrared absorption bands of water.

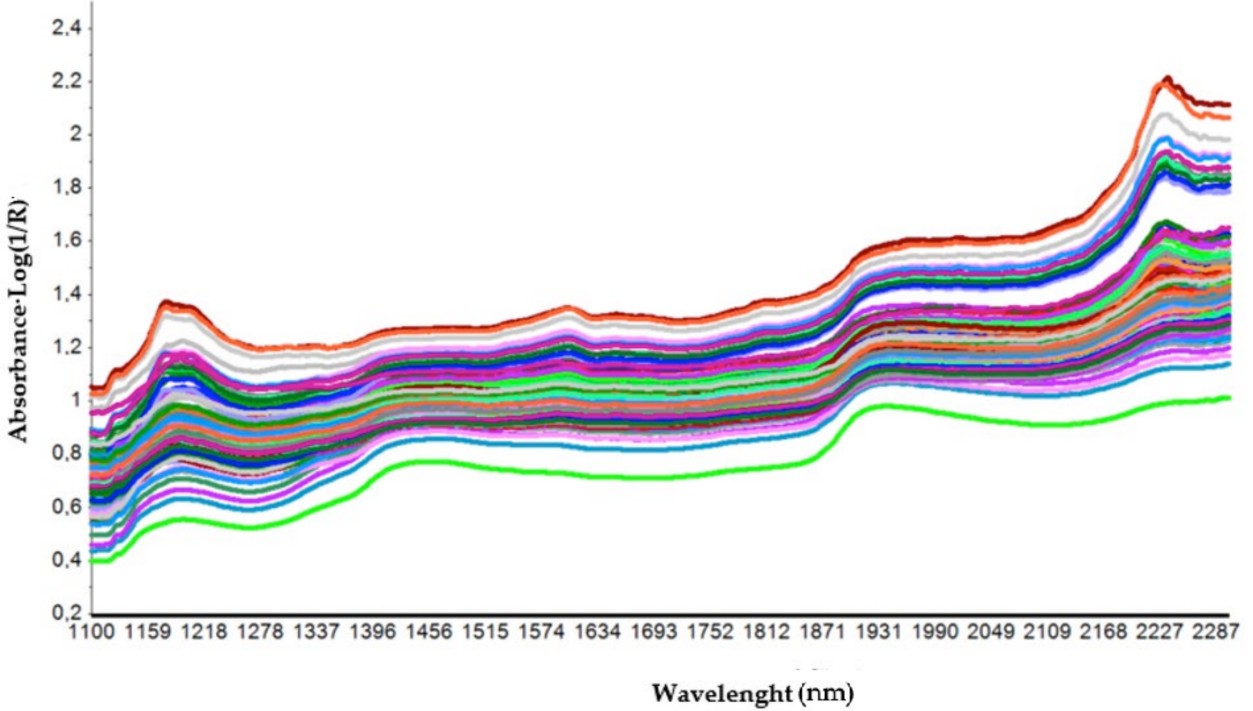

**Figure 3.** NIR raw-spectra collected with a Brimrose Luminar 5030 AOTF-NIR spectrometer during véraison until harvest, in entire grapes of 'Trincadeira', 'Aragonês', 'Syrah', and 'Touriga Nacional'.

During véraison until harvest, total phenols revealed a slow increase until harvest, except in "Aragonês", which did not reveal significant differences during this period. In both years, flavonoid content did not show an evolution pattern among varieties, increasing in some varieties along véraison and decreasing in others. On the contrary, tannins extracted from grape skins were higher in 2017 than in 2018. The variety with higher skin tannin content was "Touriga Nacional", and "Trincadeira" had the lower tannin content.

The statistics performance of the best developed prediction models is summarized in Table 3. Due to the limited number of samples, the models were validated through a leave-one-out cross-validation procedure using the Kernel algorithm. Outlier limits were calculated according to two criteria: F-residual and Hotteling's $T^2$ statistics with an imposed limit of 5%. Table 3 summarises the different statistical indicators for each model. Models were developed using raw and transformed spectra by first derivative and Standard Normal Variate procedures. The best prediction models were selected according to the values of $R^2$, RMSE, RPD, and Bias. All the selected prediction models have a $R^2$ higher than 0.85, except for total phenolic content and flavonoids, which showed an $R^2$ of 0.71 and 0.72, respectively. In fact, the prediction models for these two classes of compounds reached lower statistical indicators, denoting less robust models. The validation of the prediction models to quantify SSC, TA, total anthocyanins, and tannins reached $R^2$ values of 0.86, 0.86, 0.81, and 0.82, respectively. These models also showed *RPD* values higher than 2.3, which indicates that these can be used for rough screening [45]. In addition, the

lower *RMSE* values and the higher $R^2$ values indicate the most accurate prediction model to be selected.

**Table 3.** Statistical parameters to characterize the quality of the final models to quantify total phenolic content, flavonoids, anthocyanins, and tannins in entire grapes.

| | **Spectra Transformations** | **Calibration Model** | | | **Cross-Validation Model** | | | |
|---|---|---|---|---|---|---|---|---|
| | | **LV** | **$R^2$** | ***RMSE*** | **$R^2$** | ***RMSE*** | ***Bias*** | ***RPD*** |
| Soluble Solids Content (%) | Raw spectra * | 6 | 0.89 | 1.260 | 0.86 | 1.457 | 0.0115 | 2.6 |
| | 1st derivative | 5 | 0.88 | 1.503 | 0.82 | 1.845 | 0.0642 | 2.2 |
| | SNV | 6 | 0.88 | 1.391 | 0.83 | 1.640 | 0.0064 | 2.4 |
| Titratable acidity | Raw spectra * | 7 | 0.90 | 0.164 | 0.86 | 0.195 | −0.0027 | 2.7 |
| | 1st derivative | 5 | 0.81 | 0.253 | 0.71 | 0.312 | −0.0086 | 1.7 |
| | SNV | 6 | 0.89 | 0.171 | 0.86 | 0.197 | 0.0007 | 2.7 |
| Total Phenolic content (mg Gallic acid eq/mL grape skin extract) | Raw spectra * | 6 | 0.71 | 0.077 | 0.61 | 0.091 | −0.0001 | 1.6 |
| | 1st derivative | 6 | 0.67 | 0.098 | 0.29 | 0.141 | 0.0018 | 1.2 |
| | SNV | 7 | 0.49 | 0.118 | 0.27 | 0.143 | 0.0014 | 1.2 |
| Total Flavonoids (mg Rutin eq/mL grape skin extract) | Raw spectra * | 6 | 0.72 | 0.020 | 0.62 | 0.023 | 0.0004 | 1.6 |
| | 1st derivative | 3 | 0.36 | 0.033 | 0.24 | 0.036 | 0.0005 | 1.1 |
| | SNV | 5 | 0.43 | 0.031 | 0.28 | 0.035 | 0.0003 | 1.2 |
| Total Anthocyanins (mg Oenin-3-O-glucosides/mL grape skin extract) | Raw spectra * | 7 | 0.87 | 0.396 | 0.81 | 0.480 | 0.0083 | 2.3 |
| | 1st derivative | 6 | 0.80 | 0.495 | 0.60 | 0.717 | 0.0105 | 1.6 |
| | SNV | 7 | 0.73 | 0.579 | 0.602 | 0.712 | 0.0008 | 1.6 |
| Total Tannins (mg Tannic acid eq/mL grape skin extract) | Raw spectra | 5 | 0.81 | 0.227 | 0.769 | 0.257 | 0.0026 | 2.0 |
| | 1st derivative | 3 | 0.79 | 0.231 | 0.67 | 0.295 | 0.0005 | 1.8 |
| | SNV * | 6 | 0.88 | 0.176 | 0.82 | 0.218 | 0.0061 | 2.4 |

The selected models are identified with an asterisk.

In Figure 4 a comparison is shown between predicted values and reference values for all the compounds analysed in calibration and validation models after eliminating the outliers. The regression analysis indicates a good correlation coefficient for all the models between the reference methods and the predicted values. To have good calibration models, it is necessary to have a wide distribution of the reference values, which is observed for all compounds (Figure 4). The sampling method preformed along the véraison period from different varieties contributed to these wide distribution ranges. The accuracy of the models was verified through a y-randomization procedure and the proper performance was confirmed by the low determination coefficients obtained.

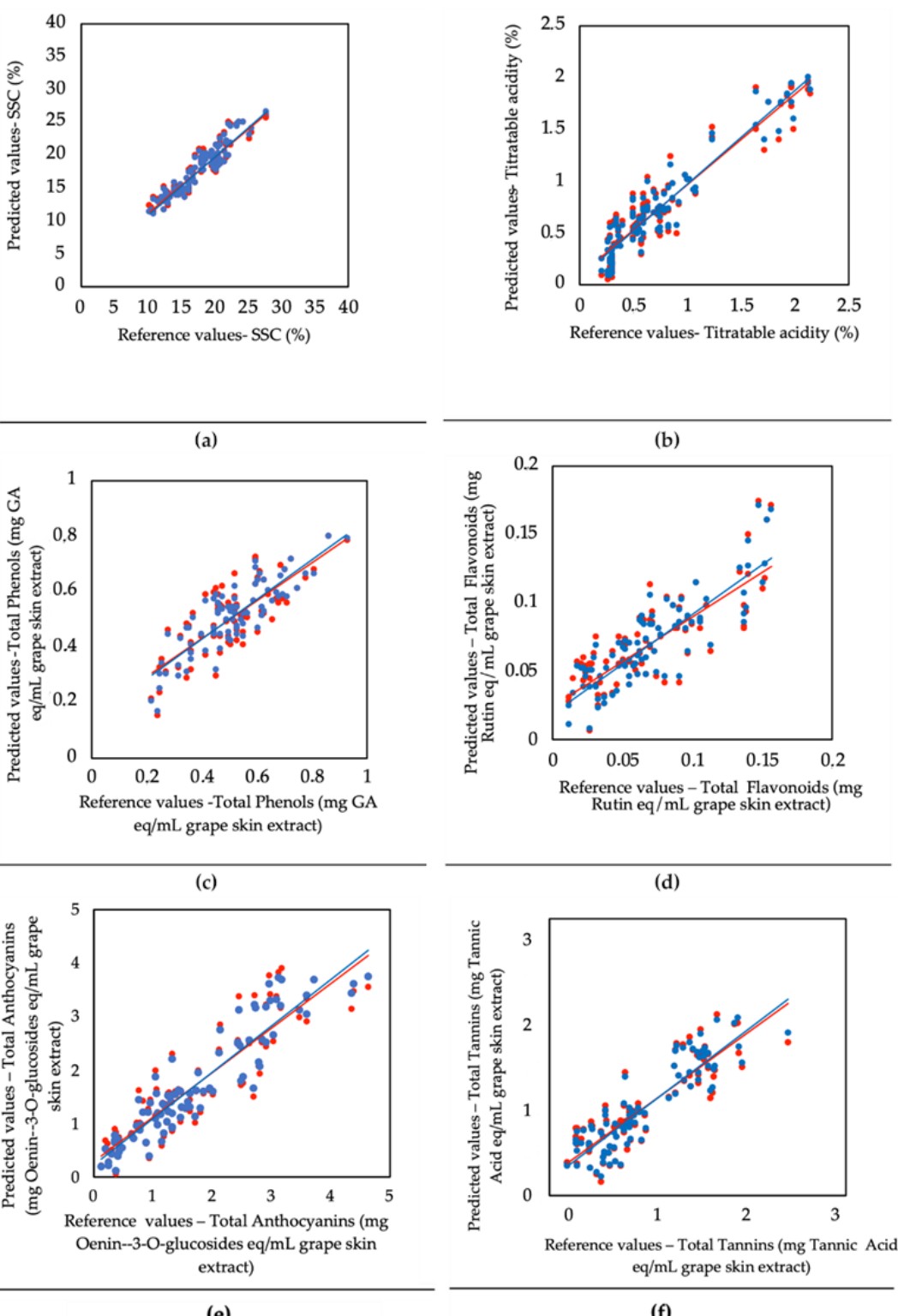

**Figure 4.** Plot of reference values vs. PLSR predicted values of calibration and cross validation sets for different compounds (**a**)—Soluble solids content -SSC (%); (**b**)—Titratable acidity (%); (**c**)—Total phenolic content (mg GA eq/mL grape skin extract); (**d**)—Total flavonoids content (mg Rutin eq/mL grape skin extract); (**e**)—Total anthocyanins (mg Oenin–3-O-glucosides eq/mL grape skin extract); (**f**)—Total of tannins (mg Tannic Acid eq/mL grape skin extract); Model developed with the calibration set; Model developed with the validation set.

## 4. Discussion

*4.1. Grape Chemical Parameters Variability*

During the ripening period that begins in véraison and ends at harvest, it is necessary to monitor grapes to access the evolution of many quality parameters in order to adopt the best management of a selective harvest. SSC, TA, phenols, flavonoids, anthocyanins, and tannins are the main compounds responsible for the high-quality characteristics of grapes. The quantification of these parameters will determine the moment of harvest.

The selected two years were very different, which justifies the differences in SSC and TA values between years. In fact, large differences in SSC may reflect variations in annual climate conditions [46]. Higher temperatures have been reported to have a negative impact on TA in many fruit species [47] and in wine grapes [48]. The effect of cold climates on the acid concentration of grapes [49] and, consequently, the effect on wine quality is widely reported. Wines produced in the northern regions of Portugal compared with the wines produced in the southern regions showed a higher TA value due to the lower temperatures in the north of Portugal [31]. SSC and TA are important quality parameters that have a great impact on wine quality and are usually used to select the right harvest date. Despite numerous factors that interact and influence SSC and TA, in both years, SSC increased during maturation, contrary to TA, which decreased during the same period. The same observations were made by Buttrose (1971) [50], who found a marked decrease in the acid concentration in grape berries. Independently of the differences between years, both TA and SSC have expected values considering the region of Portugal where the trial was located; similar values were obtained in the south region of Portugal in "Touriga Nacional" and 'Temperanillo' by Costa et al. (2020) [31].

Phenolic compounds are characterized by one or more hydroxyl groups attached to an aromatic ring. Position and number of functional groups connected to the aromatic ring will contribute to these compounds' different chemical and physical properties [51]. Most of the phenolic compounds quantified in the wine are derived primarily from the grape berry skin [1], and the most representative compounds in berry skin and seeds are flavonoids, anthocyanins, and tannins. Each of these compounds has roles of varying importance in determining the ultimate flavour and colour of the wine, which is the main reason to monitor these compounds in the vineyard during ripening. Marked differences were observed in flavonoids, anthocyanins, and tannins extracted from berry skins in 2017 and 2018, and both natural and cultural factors contributed to emphasizing those differences. Water [52], temperature, and light [53] have been referred to as important factors to be considered in canopy management that affect berry composition and, consequently, the phenolic production of the grapes [54]. Moreover, the composition of the solar spectra, mainly the increase in UV radiation, has been referred to as an important factor that contributes to the increase in flavonoids, anthocyanins, and tannins in wine grapes [53]. Despite the differences in this group of compounds, results of Total Phenolic Content showed just a slight increase along the véraison period in all grape varieties, except for 'Aragonês'. Colorimetric methods, like Folin–Ciocalteu, are widely used to quantify Total Phenolic Content, however, this is a bulk reaction and diverse compounds are sensible to this reaction, leading to over-estimations of total phenols in the grape must extracts. This over-estimation of total phenols justifies the lack of differences in these compounds between years and the marked differences observed in flavonoids, anthocyanins, and tannins extracted from berry skins in 2017 and 2018.

Total Anthocyanins Content values are in accordance with those found by Chaves et al. (2010) [55] in Alentejo, despite the small range of these values due to the increase in the dilution factor that occurs during berry development. In this study, the extraction of anthocyanins was carried out using the same skin to solvent ratio, which avoids the dilution effect. However, the same authors found higher values for anthocyanins in 'Aragonês' compared with "Touriga Nacional". In fact, the degree of ripeness, the cultural practices, the genetic potential of the variety, and the inter-annual climate variability may affect the anthocyanins' profile and concentration at harvest, which may justify the

differences between varieties and years [31]. In addition, Total Flavonoids Content was influenced by the different climate conditions in 2017 and 2018. In fact, Wilson (2001) [56] found, in *B. napus* leaves, that the flavonoids extracted from leaves decrease in response to the increase in UVA radiation due to photo-oxidation.

The values of Total Tannins Content between the years were different, being higher in 2017, and this trend was also reported by Kyraleo et al. (2017) [57], who observed differences that varied by up to double in skin total tannins within two consecutive years. These compounds are usually found in grape seeds but are also found in smaller amounts in skins [58]. Generally, the tannin content in grape skins increases during maturation in all varieties until harvest. Kennedy (2001) [59] reported that skin tannins increase in size during early stages of ripening and undergo reactions with skin cell wall pectins and anthocyanins along véraison.

### 4.2. Qualitative Analysis of Prediction Models

In routine analysis, these compounds are quantified after grapes have been harvested, and this operation is usually carried out in the laboratory. Little research has been conducted using portable devices in the field or in near-field conditions. In the last 15 years, the quantification methods for phenolic compounds have gone through major improvements, and some rapid methods with some degree of automation have been developed using FTIR devices [60]. However, these are destructive methods that cannot be assessed in the fields. The development of non-destructive methods using portable devices to quantify grape quality parameters that can be used in the vineyards will be a great improvement in wine quality control.

The results presented in this study have demonstrated the ability of NIR spectroscopy, in combination with a portable device, to quantify in entire grapes: SSC, TA, Total Phenolic Content, Total Flavonoids Content, Total Anthocyanins Content, and Total Tannins Content under near field conditions. The most limiting factor to spectra collection, in field conditions, is natural light. In this study, spectra were collected 40 mm above the sample under natural light to reproduce, as faithfully as possible, the natural field conditions. Other authors have reported the use, in field, of portable spectrometers to predict grape key parameters; however, the wavelength range of these spectrometers did not include the NIR band [61]. The wide NIR wavelength range obtained with the Brimrose portable spectrometer allowed better prediction models for anthocyanins and phenols when compared with the performance of the models obtained by Fernandez-Novales (2019) [61].

Some attempts have been made to predict phenols in field conditions using portable spectrometers ([62] and Fernandez-Novales (2019) [61]). However, the quality of the prediction models was just fairly good. Most of these models are capable of predicting only a small group of compounds with an impact on grape quality. To the extent of our knowledge, this study is the first attempt to assess six different quality parameters (SSC, TA, total phenolic compounds, total anthocyanins, total flavonoids, and total tannins content) simultaneously in intact grapes of four different varieties using a portable NIR spectrometer. All the prediction models developed have determination coefficients over 81%, except for total flavonoids and total phenolic content, which have an $R^2$ of 72% and 71%, respectively. All the RPD values in the prediction models for SSC, TA, total anthocyanins, and total tannins content were over 2.3, indicating the possibility of using these in rough screening. Despite the generally lower RPD values, the combination of these prediction models and the NIR spectrometer in motorized vehicles will much improve the process of grape sampling for quality evaluation in the vineyards during véraison. Currently, this process is carried out manually; the existence of a methodology to accelerate this process and to evaluate extended areas will overcome some problems due to sample representativeness. It is expected that the inclusion of additional data from a following year will improve the performance of these less accurate models. The findings of this study suggest optimistic expectations for future models developed with new data.

On the other hand, SSC and TA were successfully predicted using different wavelength ranges including NIR and visible regions. Due to the increase in SSC along with colour intensification all through véraison, it is possible to develop robust prediction models using the visible wavelength range [61]. The NIR wavelength range has been often used to develop prediction models for SSC and TA in many fruit species [63], and particularly in grapes [64]. In this study, the RPD values for SSC and TA prediction were the higher values, and above 2.5, which indicates a minimum good potential for quantitative predictions of the model [65].

## 5. Conclusions

To produce high-quality red wines, it is necessary to follow up the evolution of several grape compounds. SSC, TA, phenols, flavonoids, anthocyanins, and tannins are the main compounds that have a higher impact on grape quality characteristics. The development of new methodologies using NIR spectroscopy will be a great advantage over analytical procedures. This study's outcomes demonstrate that it is possible, using a NIR portable spectrometer (Brimrose Luminar 5030), to predict in situ the SSC, TA, anthocyanins, and tannins with a relatively good accuracy. The prediction models for flavonoids and phenols indicate a minimum acceptable potential for quantitative predictions. In spite of the lower accuracy of these models, the possibility of quantifying in situ several grape compounds in four different grape varieties simultaneously is an important achievement in grape quality evaluation. The combination of portable spectrometers and motorized vehicles will allow more information to be collected in larger areas, overcoming the problem of representativeness in manual grape sampling.

**Supplementary Materials:** The following supporting information can be downloaded at: https://www.mdpi.com/article/10.3390/agronomy12030637/s1, Figure S1. Post-Hoc Tukey HSD test from a factorial ANOVA for the Total Phenolic content (mg GA eq/ml grape skin extract) among grape varieties in 2017 and 2018 years. Bars that do not share a letter are significantly different with a $p < 0.05$. Figure S2. Post-Hoc Tukey HSD test from a factorial ANOVA for Total Flavonoids (mg Rutin eq/ml grape skin) among grape varieties in 2017 and 2018 years. Bars that do not share a letter are significantly different with a $p < 0.05$. Figure S3. Post-Hoc Tukey HSD test from a factorial ANOVA for the Total Anthocyanins (mg Oenin-3-O-glucosides/ml grape skin extract among grape varieties in 2017 and 2018 years. Bars that do not share a letter are significantly different with a $p < 0.05$. Figure S4. Post-Hoc Tukey HSD test from a factorial ANOVA for Total of Tanins (mg Tannic acid eq/ml grape skin extract) among grape varieties in 2017 and 2018 years. Bars that do not share a letter are significantly different with a $p < 0.05$. Figure S5. Post-Hoc Tukey HSD test from a factorial ANOVA for Soluble Solids Content (%) among grape varieties in 2017 and 2018 years. Bars that do not share a letter are significantly different with a $p < 0.05$. Figure S6. Post-Hoc Tukey HSD test from a factorial ANOVA for Titratable Acidity (%) among grape varieties in 2017 and 2018 years. Bars that do not share a letter are significantly different with a $p < 0.05$.

**Author Contributions:** Conceptualization, A.E.R., M.R.M. and J.M.B.; methodology, A.E.R., M.I.R., G.C.M. and M.R.M.; writing—original draft preparation M.I.R., M.R.M. and A.E.R.; writing—review and editing, A.E.R., M.R.M. and J.M.B.; data processing and statistical analysis, A.E.R., and M.R.M.; project administration, A.E.R. and M.R.M.; funding acquisition, A.E.R., M.I.R. and M.R.M. All authors have read and agreed to the published version of the manuscript.

**Funding:** This work was funded by National Funds through FCT (Foundation for Science and Technology) under the Project UIDB/05183/2020.

**Institutional Review Board Statement:** Not applicable.

**Informed Consent Statement:** Not applicable.

**Data Availability Statement:** Not applicable.

**Acknowledgments:** Authors acknowledge the financial support provided by FEDER and National Funds through the Programa Operacional Regional ALENTEJO 2020—QualFastNut—"Utilização da espectroscopia NIR para a análise rápida da qualidade em frutos secos". The authors thank the Projects UIDB/05183/2020, UIDB/04449/2020, and UIDP/04449/2020, funded by FCT (Foundation for Science and Technology).

**Conflicts of Interest:** The authors declare no conflict of interest.

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
