# Peer review of "Quality Assessment of Red Wine Grapes through NIR Spectroscopy"

_agronomy, doi:10.3390/agronomy12030637_

Round 1

Reviewer 1 Report

The manuscript "Quality assessment of red wine grapes through NIR spectroscopy" is one with basic analysis. But it is interesting to correlate with NIR.
In the abstract the results should be presented more clearly and in detail.
The purpose of this study is missing from the abstract.
In the abstract, abbreviations for NIR and RPD are used. Given that it is the first mention in the text, recalling the explanation of the abbreviations.
The introduction lacks concrete data on existing studies in literature.
What does PLS mean?
Statistical analysis of the data presented in Table 1, Table 2 is missing. The standard deviation is not sufficient
Table 2 is very crowded. I recommend after performing the statistical analysis to be done horizontally.

Author Response

Thank you for reviewing our manuscript, we are grateful for all the comments which were valuable improvements to our paper. We have incorporated most of the suggestions. Those changes are highlighted within the manuscript. Please see  the attached document.

Reviewer 2 Report

The Abstract of the Manuscript contains more than 200 words. It should be adjusted according to the Instructions for the Authors.

Manuscript contains all the elements requested by the Journal.

In References no. 5 and 7 there are no given Publishers.

Grammar needs to be revised.

Author Response

To the reviewer,

Thank you for reviewing our manuscript, we are grateful for all the comments which were valuable improvements to our paper. We have incorporated most of the suggestions. Those changes are highlighted within the manuscript. Please see  the attached document

Reviewer 3 Report

The authors have carried out a detailed characterization of red wine grapes based on NIR spectroscopy and PLS regression. They have focused more on the phenolic compounds of the samples. In my opinion the experimental part of the study was adequate, and they covered a wide range of the characterization of the samples. The introduction part was also well-written. However, they have missed several points, where the models could be improved and validated properly.

My major problems with the manuscript are as follows:

  • The authors have discussed NIR spectroscopy and PLS regression in the introduction part, but it should be transferred to the materials and methods section. On the other hand, I have really missed the presentation of those earlier models from the literature, which have used the same methodology (NIR) or predicted the same quantitative parameters.
  • Even if there were a limited number of samples, it would have been prudent to randomly select 10 as external set in the beginning of the analysis and use 5-fold instead of the 20-fold CV. If the authors decided to use only CV in this way, at least they should iterate the process in several rounds to see the variance of the errors in the validation step.
  • The authors have provided lots of tables with the reference measurements, but they have not provided any statistical analysis of them, which could detect significant differences between the grape varieties or the year. I suggest using virtually any basic statistical tool, such as a t-test or nonparametric tests, maybe ANOVA. It would be nice to see the tendencies in graphs instead of the raw numbers.
  • I really miss that the authors did not try other regression methods, such as SVM for the analysis. Please try it, it can improve the models and it is available even in Unscrambler. Also, the use of any variable selection tool is missing. Please try the most common ones, such as interval PLS, VIP scores, or genetic algorithm. It can be also a nice improvement for the models.
  • In the validation phase of the models, the authors should use randomization test (X or Y scrambling) to verify the goodness of the models. It is also a common procedure, and it can be easily done manually with the mixing of the X or Y variables.
  • It would be great to compare the performance of the models with the benchmark models (literature models) in a table format.

Minor comments:

  • The tables and Figure 4 have some editing issue. Sometimes the lines are not in the proper places and in the figure, there are disturbing white squares and strange text parts which can cover the text in the y axis.
  • In Figure 3, please write “nm” after the title of the X axis in bracket.
  • “RMSE indicates the accuracy in which a sample can be predicted and is comparable to a RMSECV when a cross-validation procedure is used instead using a separate sample test.” – please use “error” instead of “accuracy”, because the meaning is the opposite.

In summary, the manuscript has a great value and potential interest, but after the careful chemical measurements, unfortunately the modeling part is poorly described in the present form of the text. I recommend a major revision by extending the regression modeling part according to the above.

Author Response

(The authors gave the same response as above.)

Round 2

Reviewer 1 Report

The authors responded to all my comments.

Reviewer 3 Report

The manuscript has improved a lot, but some minor changes are still needed:
-    In Figure 3, “nm” should be next to the title of the X axis (Wavelength (nm)), not in the title of the figure.
-    Please provide some results about the y-randomization procedure, at least in the supplementary material.
-    I am still missing a simple table, comparing the final models in the end of the manuscript to the benchmark studies from the literature.
-    row 525: 2.3 instead of 2,3, please check the text for potential typos.